# Enmein Decreases Synaptic Glutamate Release and Protects against Kainic Acid-Induced Brain Injury in Rats

**DOI:** 10.3390/ijms222312966

**Published:** 2021-11-30

**Authors:** Cheng-Wei Lu, Yu-Chen Huang, Kuan-Ming Chiu, Ming-Yi Lee, Tzu-Yu Lin, Su-Jane Wang

**Affiliations:** 1Department of Anesthesiology, Far-Eastern Memorial Hospital, New Taipei City 22060, Taiwan; drluchengwei@gmail.com (C.-W.L.); yuchenhuang@msn.com (Y.-C.H.); 2Department of Mechanical Engineering, Yuan Ze University, Taoyuan 32003, Taiwan; 3Division of Cardiovascular Surgery, Cardiovascular Center, Far-Eastern Memorial Hospital, New Taipei 22060, Taiwan; chiu9101018@gmail.com (K.-M.C.); mingyi.lee@gmail.com (M.-Y.L.); 4Department of Nursing, Asia Eastern University of Science and Technology, New Taipei City 22060, Taiwan; 5Department of Photonics Engineering, Yuan Ze University, Taoyuan 32003, Taiwan; 6Research Center for Chinese Herbal Medicine, College of Human Ecology, Chang Gung University of Science and Technology, Taoyuan 33303, Taiwan; 7School of Medicine, Fu Jen Catholic University, New Taipei City 24205, Taiwan

**Keywords:** enmein, glutamate release, Ca^2+^ influx, glutamate transporter, neuroprotection, antiinflammation, excitotoxicity

## Abstract

This study investigated the effects of enmein, an active constituent of *Isodon japonicus* Hara, on glutamate release in rat cerebrocortical nerve terminals (synaptosomes) and evaluated its neuroprotective potential in a rat model of kainic acid (KA)-induced glutamate excitotoxicity. Enmein inhibited depolarization-induced glutamate release, FM1-43 release, and Ca^2+^ elevation in cortical nerve terminals but had no effect on the membrane potential. Removing extracellular Ca^2+^ and blocking vesicular glutamate transporters, N- and P/Q-type Ca^2+^ channels, or protein kinase C (PKC) prevented the inhibition of glutamate release by enmein. Enmein also decreased the phosphorylation of PKC, PKC-α, and myristoylated alanine-rich C kinase substrates in synaptosomes. In the KA rat model, intraperitoneal administration of enmein 30 min before intraperitoneal injection of KA reduced neuronal cell death, glial cell activation, and glutamate elevation in the hippocampus. Furthermore, in the hippocampi of KA rats, enmein increased the expression of synaptic markers (synaptophysin and postsynaptic density protein 95) and excitatory amino acid transporters 2 and 3, which are responsible for glutamate clearance, whereas enmein decreased the expression of glial fibrillary acidic protein (GFAP) and CD11b. These results indicate that enmein not only inhibited glutamate release from cortical synaptosomes by suppressing Ca^2+^ influx and PKC but also increased KA-induced hippocampal neuronal death by suppressing gliosis and decreasing glutamate levels by increasing glutamate uptake.

## 1. Introduction

Glutamate is an excitatory neurotransmitter that is involved in many physiological functions of the brain [1,2]. However, excess glutamate in the synaptic cleft causes an overactivation of glutamate receptors, which increases the intracellular Ca^2+^ concentration and results in reactive oxygen species generation, oxidative stress, and mitochondrial dysfunction, ultimately resulting in neuronal cell damage or death. In addition, changes in glutamate levels can disrupt glutamatergic neurotransmission, which can be a cause of neurological diseases such as stroke, epilepsy, traumatic brain injury, and psychiatric and neurodegenerative diseases [3,4]. Therefore, preventing glutamate excitotoxicity in the nervous system is considered to be a promising target for clinical trials to develop drugs to treat these neurological disorders [5,6,7].

Natural products are an invaluable source for novel drug research because of their safety and various biological functions [8]. The search for and discovery of active components in medicinal plants is a popular topic in pharmaceutical chemistry. In particular, many natural constituents from medicinal plants have been reported to reduce synaptic glutamate release and thereby protect neurons, including baicalein, a flavonoid from Scutellaria baicalensis; echinacoside, a phenylethanoid glycoside from *Herba Cistanche*; 11-keto-*β*-boswellic acid, a triterpenoid from *Boswellia serrata*; and asiatic acid, a triterpene from *Centella asiatica* [9,10,11,12,13]. Therefore, natural products that inhibit central glutamatergic neurotransmission may offer a possible therapeutic option for treating neurological diseases [13]. Enmein is a diterpenoid derived from the plant *Isodon japonicus* Hara that is widely used in dietary supplements and folk medicine [14]. Enmein has antibacterial [15], anti-inflammatory [16], antitumor [17], and immunosuppressive effects [18]. There is currently no data to support the effects of enmein on the central glutamate system. The aim of the present study was to evaluate: (1) whether enmein has an effect on glutamate release in the cerebral cortex nerve terminals (synaptosomes) of rats; and (2) whether enmein protects against excitotoxicity caused by kainic acid (KA), a glutamate analog [18]. In the synaptosomal model, the effects of enmein on glutamate release, membrane potential, Ca^2+^ concentrations, and the Ca^2+^-activated protein kinase C (PKC) signaling cascade were evaluated. In the KA rat model of excitotoxicity, we evaluated the effects of enmein pretreatment on neuronal damage, glutamate elevation, glial cell activation, and the expression of excitatory amino acid transporters (EAATs) 2 and 3, which are responsible for glutamate clearance, synaptic plasticity-related proteins, synaptophysin and postsynaptic density protein 95 (PSD95), and hippocampal glial fibrillary acidic protein (GFAP) and CD11b.

## 2. Results

### 2.1. The Influence of Enmein on Glutamate Release in Rat Cerebrocortical Nerve Terminals

To investigate the influence of enmein on glutamate release, isolated nerve terminals were depolarized with the K^+^ channel blocker 4-Aminopyridine (4-AP), which increases Ca^2+^ influx and glutamate release [19]. Figure 1A shows that in synaptosomes incubated in the presence of 1 mM CaCl_2_, 4-AP (1 mM) evoked significant glutamate release. Incubation with enmein (30 μM) for 10 min before 4-AP administration did not have a significant effect on the basal release of glutamate but markedly reduced the glutamate release mediated by 4-AP [t(8) = 3.9, *p* < 0.001 vs. the control group]. Enmein inhibited glutamate release concentration-dependently, with a maximum inhibition of 56.3% ± 1.6% produced at 30 μM and a half maximal effective concentration of 8 μM (Figure 1B). In addition, in extracellular Ca^2+^-free medium containing EGTA, glutamate release evoked by 4-AP from synaptosomes was reduced to approximately 67% of that detected in the presence of external Ca^2+^ [t(8) = 0.8, *p* < 0.001]. Notably, this 4-AP-evoked Ca^2+^-independent release was not affected by prior enmein addition (*p* = 0.24 vs. Ca^2+^-free group; Figure 1C). Similarly, enmein failed to cause significant inhibition in the presence of bafilomycin A1, a vacuolar H^+^ ATPase inhibitor [F(3,16) = 1467.1; *p* = 0.16 vs. bafilomycin A1-treated group, Figure 1C, inset]. Bafilomycin A1 alone did reduce 4-AP-evoked glutamate release (*p* < 0.001). Furthermore, the effect of enmein on 4-AP-triggered exocytosis was evaluated by analyzing the reduction in FM1-43 fluorescence intensity [20]. As shown in Figure 1D, 4-AP (1 mM) caused a decrease in FM1-43 fluorescence in the presence of CaCl_2_. Preincubating synaptosomes with enmein for 10 min significantly reduced the 4-AP-induced decrease in FM1-43 fluorescence [t(8) = 6.3; *p* < 0.001]. These results suggest that the inhibition of glutamate release by enmein is due to a decrease in the Ca^2+^-dependent exocytotic mechanism.

### 2.2. The Influence of Enmein on Cytosolic Free Ca^2+^ Concentration ([Ca^2+^]_C_) and Synaptosomal Plasma Membrane Potential

As shown in Figure 2A, fluorometric experiments performed in synaptosomes containing Fura-2-acetoxymethyl ester (Fura-2-AM) and depolarized with 1 mM 4-AP showed that the rise of [Ca^2+^]_C_ above basal levels was significantly reduced in the presence of enmein [t(8) = 9.1, *p* < 0.001 vs. control group]. However, enmein had no significant effect on basal [Ca^2+^]_C_ levels [t(8) = 1.4, *p* = 0.21 vs. control group]. In addition, we examined the effect of enmein on synaptosomal plasma membrane potential by using the membrane potential-sensitive dye DiSC_3_(5). As shown in Figure 2B, 4-AP (1 mM) caused an increase in DiSC_3_(5) fluorescence. Preincubation of synaptosomes with enmein had no effect on the resting plasma membrane potential [t(8) = 0.25, *p* = 0.8 vs. control group]; however, the 4-AP-induced elevation of DiSC_3_(5) fluorescence was significantly decreased in the presence of enmein [t(8) = 1.8, *p* = 0.1 vs. control group].

### 2.3. The Reduction in N- and P/Q-type Ca^2+^ Channel Activities Is Involved in the Enmein-Induced Inhibition of Glutamate Release

In rat cerebrocortical nerve terminal preparations, N- and P/Q-type voltage-dependent Ca^2+^ channels (VDCCs) promoted glutamate release [21]. Similarly, when VDCCs were blocked with 2 μM ω-conotoxin GVIA (ω-CgTX GVIA) (N-type) or 200 nM ω-agatoxin IVA (ω-AgTX IVA) (P/Q-type), 4-AP-evoked glutamate release was largely reduced [F(3,23) = 461.3; *p* < 0.001 vs. control group], and the inhibitory effect of enmein was not detectable (*p* = 0.99 vs. ω-CgTX GVIA and ω-AgTX IVA-treated group; Figure 3). Similar results were obtained using 1 μM ω-CgTX MVIIC, a wide-spectrum blocker of N-, P-, and Q-type Ca^2+^ channels. ω-CgTX MVIIC alone reduced 4-AP-evoked glutamate release [F(3,23) = 381.8; *p* < 0.001 vs. control group]. However, the release measured in the presence of both ω-CgTX MVIIC and enmein was similar to that obtained in the presence of only ω-CgTX MVIIC *(p* = 0.47). In addition, we examined the effect of enmein on 15 mM KCl-evoked glutamate release, a mechanism that involves Ca^2+^ influx primarily via VDCC openings [22]. As shown in Figure 3, enmein also significantly inhibited KCl-evoked glutamate release [t(8) = 17.9, *p* < 0.001 vs. control].

### 2.4. The Suppression of PKC/MARCKS Pathways Is Linked to Enmein-Induced Inhibition of Glutamate Release

Increased Ca^2+^ influx in nerve terminals has been shown to enhance PKC activation and subsequently enhance glutamate release [23]. We therefore investigated whether the inhibition of Ca^2+^ influx caused by enmein decreased PKC activity and thus decreased glutamate release. As shown in Figure 4A, 10 μM isindolylmaleimide I (GF109203X), a general PKC inhibitor, reduced 4-AP-evoked glutamate release [F(3,25) = 491.7, *p* < 0.001 vs. control group]. When GF109203X was present, enmein did not significantly inhibit glutamate release (*p* = 0.98 vs. GF109203X-treated group). Similarly, in the presence of 1 μM, 5,6,7,13-tetrahydro-13-methyl-5-oxo-12H-indolo[2,3-a]pyrrolo[3,4-c]carbazole-12-propanenitrile (Go6976), an inhibitor of conventional Ca^2+^-dependent PKC-α isozymes, 4-AP-evoked glutamate release was reduced [F(3,23) = 465.9; *p* < 0.001 vs. control group]. In addition, enmein did not reduce 4-AP-evoked glutamate release when Go6976 was present (*p* = 0.99 vs. Go6976-treated group; Figure 4A). Furthermore, Figure 4B shows that synaptosome depolarization with 1 mM 4-AP in the presence of 1.2 mM CaCl_2_ increased the phosphorylation of PKC [F(2,12) = 874.5; *p* < 0.001 vs. control group], PKC-α [F(2,12) = 39.9; *p* < 0.001 vs. control group], and myristoylated alanine-rich C kinase substrate (MARCKS), a key presynaptic substrate for PKC [F(2,12) = 142.1; *p* < 0.001 vs. control group]. These 4-AP-induced phosphorylation elevations were markedly decreased by the presence of enmein (*p* < 0.05 vs. the 4-AP-treated group). 4-AP did not affect PKC and PKC-α expression [PKC, F(2,12) = 0.44; *p* = 0.95 vs. control group; PKC-α, F(2,12) = 0.69; *p* = 0.99 vs. control group] (Figure 4B,C).

### 2.5. The Influence of Enmein on Neuronal Death in the Hippocampi of KA Rats

The neuroprotective effect of enmein was evaluated in a rat model of KA-induced glutamate excitotoxicity. As shown in Figure 5A, rats were given either enmein or dimethylsulfoxide (DMSO) intraperitoneally (i.p.) 30 min before KA injection (i.p.). Nissl staining was performed to examine neuronal survival in the hippocampus 72 h after KA injection. Compared to the control group, in the KA-treated group, significant neuronal loss in the CA1 and CA3 regions was observed [CA1, F(2,6) = 483.9, *p* < 0.001; CA3, F(2,6) = 279.4, *p* < 0.001; Figure 5B]. However, in the enmein/KA-treated group, the number of surviving neurons in the CA1 and CA3 regions was significantly higher than that in the KA-treated group (*p* < 0.05; Figure 5B). In addition, the histofluorescence findings of neuronal nuclear protein (NeuN) (a specific neuron marker) were consistent with those of Nissl staining. Compared to that in the control group, the number of NeuN-positive neurons in the KA-treated group were significantly decreased in both the CA1 and CA3 regions [CA1, F(2,6) = 684.3, *p* < 0.001; CA3, F(2,6) = 642.9, *p* < 0.001]. Compared to that in the KA-treated group, the number of NeuN-positive neurons in the enmein/KA-treated group was increased in both the CA1 and CA3 regions (*p* < 0.05; Figure 5A,C). Similarly, the expression of NeuN in the hippocampus was significantly lower in the KA-treated group than in the control group [F(2,9) = 318.7, *p* < 0.001]. In contrast, the expression of NeuN in the hippocampus was higher in the enmein/KA-treated group than in the KA-treated group (*p* < 0.05; Figure 5D).

### 2.6. The Influence of Enmein on Glutamate Elevation and the Expression of EAAT2 and EAAT3 in the Hippocampi of KA Rats

To elucidate the possible mechanism of the neuroprotective action of enmein in our study, glutamate concentrations—a key neurotransmitter in the central nervous system underlying the pathophysiology of neurological disorders—were determined 72 h after KA injection by high-performance liquid chromatography (HPLC). As shown in Figure 6A, glutamate concentrations in the hippocampi of the KA-treated group were significantly higher than in the control group [F(2,9) = 164.7, *p* < 0.001]. In contrast, the hippocampal glutamate level of the enmein/KA-treated group was significantly lower than that of the KA-treated group (*p* < 0.05; Figure 6A). Excitatory amino acid transporters (EAATs), especially EAAT2 and EAAT3, are critical for maintaining low extracellular glutamate concentrations and preventing excitotoxicity [24,25,26]. To investigate how enmein reduced KA-induced glutamate elevation in the hippocampus, we examined the expression of EAAT2 and EAAT3 in the hippocampus. As shown in Figure 6B, Western blot analysis showed that EAAT2 and EAAT3 hippocampal levels were significantly lower in the KA-treated group than in the control group [EAAT2, F(2,9) = 12.9, *p* < 0.01; EAAT3, [F(2,9) = 30.5, *p* < 0.001]. However, in the enmein/KA-treated group, the expression levels of EAAT2 and EAAT3 were significantly higher than those in the KA-treated group (*p* < 0.05) (Figure 6B).

### 2.7. The Influence of Enmein on the Expression of Synaptic Marker Proteins (Synaptophysin and PSD95) in the Hippocampi of KA Rats

KA-induced neuronal toxicity has been shown to cause synaptic dysfunction [27,28]. The hippocampal levels of the synaptic marker proteins synaptophysin and PSD95 were evaluated 72 h after KA injection. As shown in Figure 7, Western blot analysis revealed that synaptophysin and PSD95 levels in the hippocampus were lower in the KA-treated group than in the control group [synaptophysin, F(2,9) = 106.6, *p* < 0.001; PSD95, F(2,9) = 270.3, *p* < 0.001]. Animals pretreated with enmein had significantly higher levels than the KA-treated group (*p* < 0.05).

### 2.8. The Influence of Enmein on the Activation of Glial Cells in the Hippocampi of KA Rats

Accumulating evidence has suggested that KA-induced excitotoxicity enhances glial cell activation and promotes neuroinflammatory responses in the brain [29,30,31]. As an important neuroinflammation mediator, we analyzed the expression of GFAP (an astrocyte marker) and CD11b (a microglial marker) proteins in the hippocampal CA1 and CA3 regions of the experimental animals 72 h after KA injection. Compared to the control group, the KA-injected group displayed a significant induction of gliosis, as revealed by the increased expression of GFAP and CD11b proteins [GFAP, F(2,9) = 68.7, *p* < 0.001; CD11b, F(2,9) = 99.6, *p* < 0.001] (Figure 8A). Immunohistochemical analysis further confirmed the induction of gliosis in the brain slices of KA-injected rats, as illustrated by a higher number of GFAP-positive [CA1, F(2,6) = 254.5, *p* < 0.001; CA3, F(2,6) = 182.4, *p* < 0.001] and Iba-1-positive cells [CA1, F(2,6) = 24.1, *p* < 0.01; CA3, F(2,6) = 74.3, *p* < 0.001] in the hippocampal CA1 and CA3 regions than in the control group (Figure 8B–D). In contrast, the expression of GFAP and Iba-1 in the hippocampus was lower in the enmein/KA-treated group than in the KA-treated group (*p* < 0.05; Figure 8A). The brain slices of the enmein/KA-treated group also had a reduced number of GFAP- and Iba-1-positive cells in the hippocampal CA1 and CA3 regions (*p* < 0.05 vs. KA group; Figure 8B–D). These findings show that KA-induced excitotoxic damage is related to neuroinflammation and that pretreatment with enmein could protect the brain from the neuroinflammatory mediator associated with the pathological state.

## 3. Discussion

Excess glutamate is one of the main causes of neuronal death in neurological diseases [4]. Research on new drugs capable of reducing glutamate toxicity has received increasing attention. Among these, natural constituents from medicinal plants have been suggested to regulate the glutamate system and act as promising treatments for neurological disorders [13]. Diterpenoids are important and widely distributed natural compounds with various biological effects, including antitumor, anti-inflammatory, and immune modulation effects [32]. Enmein is a diterpenoid extracted from *Isodon japonicus* Hara that exhibits antitumor, anti-inflammatory, and immune modulation effects [16,17,33]. The present study provides the first description of the role of enmein in the central glutamate system, including its ability to regulate glutamate release from rat cerebrocortical nerve terminals and exhibit neuroprotective effects in a rat model of KA-induced glutamate excitotoxicity.

### 3.1. Inhibition of Presynaptic Glutamate Release by Enmein

The first relevant finding of this study was that enmein was able to inhibit the release of glutamate evoked by 4-AP depolarization of rat cerebrocortical nerve terminals. Concerning the cellular mechanism enmein-mediated inhibition of glutamate release, we considered two possibilities: (1) alteration of nerve terminal excitability and downstream regulation of Ca^2+^ influx into the terminal; and (2) direct regulation of exocytosis-coupled VDCCs that affect Ca^2+^ influx [19,34]. However, the first possibility was ruled out due to three observations. First, in the presence of Ca^2+^-free medium containing EGTA, enmein failed to significantly inhibit 4-AP-evoked glutamate release (Ca^2+^-independent release), which depends solely on the membrane potential [35]. This indicates that enmein affects glutamate release by reducing 4-AP-evoked classical external Ca^2+^-dependent exocytosis. The vacuolar H^+^ ATPase inhibitor bafilomycin A1, which causes glutamate depletion in synaptic vesicles, completely abolishes the inhibitory effect of enmein on 4-AP-evoked glutamate release, which supports this suggestion. Second, enmein had no effect the synaptosomal membrane potential, which was measured with a membrane potential-sensitive dye. Third, the inhibitory effect of enmein was observed when KCl was used as a depolarizing agent, indicating that the involvement of K^+^ channels is not required [19]. These findings suggest that the inhibition of glutamate release by enmein is not a result of suppressing synaptosomal excitability. In addition, we demonstrated that enmein reduced the 4-AP-evoked increase in [Ca^2+^]_C_ using Fura-2. Moreover, the inhibitory effect of enmein on 4-AP-evoked glutamate release was completely prevented when release-coupled N- and P/Q-type Ca^2+^ channels were blocked. From the above results, we inferred that enmein suppresses evoked glutamate release through direct suppression of N- and P/Q-type Ca^2+^ channels. However, whether enmein has a direct interaction with presynaptic VDCCs remains unknown. On the other hand, increased cytosolic Ca^2+^ in synaptic terminals can activate PKC to phosphorylate several proteins involved in exocytosis, including synapsin I. The phosphorylation of synapsin I causes cytoskeletal disassembly, which increases synaptic vesicle availability and glutamate release [23,36]. In the present study, we also found that the inhibitory action of enmein on 4-AP-evoked glutamate release was blocked by inhibiting PKC with the specific inhibitor GF109203X or Go6976. Furthermore, enmein significantly suppressed 4-AP-evoked PKC and its substrate synapsin I phosphorylation. These results, combined with the possibility that enmein effectively suppresses glutamate release components supported by N- and P/Q-type Ca^2+^ channels, suggest that enmein decreases Ca^2+^ influx through N- and P/Q-type Ca^2+^ channels, thereby reducing PKC/synapsin I phosphorylation and, as a result, reducing glutamate release from cerebrocortical synaptosomes.

### 3.2. Prevention of Glutamate Excitotoxicity by Enmein

The second relevant finding of this study was that enmein had a neuroprotective effect in a KA-induced glutamate excitotoxicity rat model. KA is a glutamate analog, and its neurotoxicity largely depends on enhancing the glutamate system, most likely through increasing glutamate receptor activation and glutamate release [37,38]. The systemic administration of KA in rats induces neuronal damage and gliosis in various brain regions, including the hippocampus, and these pathological phenomena resemble those seen in some neurological diseases [39]. Thus, KA is the most commonly used agent in animal models to test drugs for their potential neuroprotective properties [18]. In the present study, KA administration caused substantial neuronal death in the CA1 and CA3 regions of the hippocampus, as has been reported previously [37,40]. The administration of enmein before KA injection reduced KA-induced hippocampal neuronal death. The neuroprotective effect of enmein against KA-induced hippocampal damage could be linked to its effect on glutamate neurotransmission. Two lines of evidence support this notion. First, enmein pretreatment prevented KA-induced increases in glutamate levels in the hippocampus. Second, KA decreased the expression of EAAT2 and EAAT3 in the hippocampus; these effects were reversed by enmein pretreatment. EAAT2 and EAAT3 are glutamate transporters that take up excessive glutamate in the synaptic cleft and maintain glutamate homeostasis at excitatory synapses [24,41,42]. EAAT2 is highly expressed on astrocytes and accounts for more than 90% of brain glutamate uptake, whereas EAAT3 is mainly expressed on postsynaptic neurons throughout the brain [43]. Malfunction or aberrant expression of these glutamate transporters can result in excessive glutamate buildup, leading to excitotoxicity of neurons and involvement in the development of various neurological disorders [26,44,45]. Upregulation of EAAT2 and EAAT3 has been shown to suppress glutamate accumulation and limit excitotoxicity [46,47]. Thus, we infer that reduced glutamate concentrations caused by increasing glutamate uptake via glutamate transporters may explain enmein’s neuroprotection against KA-induced excitotoxicity. In addition to increasing glutamate uptake via glutamate transporters, enmein’s ability to block glutamate release from nerve terminals may help to explain its neuroprotective mechanism against KA-induced excitotoxicity in vivo. This hypothesis is based on the association between KA-induced neurotoxicity and excessive glutamate release [37,38].

In addition, glutamate-induced neurotoxicity induces synaptic dysfunction, which leads to important neuropathological effects that have a considerable impact brain functionality and behavior [48]. In the present study, KA treatment in rats reduced the levels of synaptic marker proteins (synaptophysin and PSD95) in the hippocampus, and pretreatment with enmein reversed these effects. These proteins are essential for maintaining synaptic activity and plasticity [49]. Decreases in these proteins have been linked to KA-induced excitotoxic injury [12,27,28]. Therefore, we speculate that enmein could protect against KA-induced neuronal death by maintaining synaptophysin and PSD95 protein expression in the hippocampus.

Glutamate-induced excitotoxicity is also thought to be related to inflammatory processes [48]. Neuronal death and glutamate spillover from the synaptic region can both contribute to neuroinflammation [50,51]. Neuroinflammation, including the activation of microglia and astrocytes, is often associated with KA-induced excitotoxic injury [29,30,31]. Activated glial cells increase the production of proinflammatory cytokines and chemokines, which influence glutamate homeostasis by downregulating EAAT expression and upregulating glutamate receptors, thus increasing glutamatergic neurotransmission and excitotoxicity [50,52]. This evidence suggests that controlling KA-induced neuroinflammation is vital for protecting hippocampal neurons [30]. In the present study, we observed that KA substantially increased the levels of GFAP (an astrocytic marker) and CD11b (a microglial marker), as well as the number of activated microglia and astrocytes in the hippocampus; these increases were suppressed by enmein pretreatment. Thus, the suppression of neuroinflammation could be another mechanism that explains the neuroprotection exerted by enmein against KA-induced excitotoxicity. Our findings are consistent with those of previous studies that have reported the anti-inflammatory properties of enmein [16]. Accumulated evidence from animal models suggests that anti-inflammatory drugs may have potential neuroprotective effects for several neurological diseases. Although the mechanism by which enmein suppresses glial activation was not investigated in this study, toll-like receptors (TLRs) have been shown to play a critical role in glial cell activation and subsequent hippocampal neuron excitotoxicity [53]. Furthermore, preventing TLR activation suppresses glial activation and protects against excitotoxicity [53,54]. Whether enmein suppresses the activation of TLRs, thereby reducing glial cell activation and further suppressing KA-induced excitotoxic injuries, should be investigated in future studies. Taken together, our in vivo findings confirmed that enmein pretreatment improved hippocampal neuron protection in a rat model of excitotoxicity and suggested that the mechanism involves decreased glutamate levels, increased expression of EAAT2, EAAT3, synaptophysin and PSD95, and decreased glial cell activation.

## 4. Materials and Methods

### 4.1. Animals

Sprague–Dawley male rats (150−200 g) were kept in a controlled environment (12:12 h light-dark cycle, 24 ± 1 °C, 55% relative humidity) with free access to food and water. All experimental protocols were designed with minimal animals. Experiments and animal use procedures were carried out in accordance with the National Institutes of Health Guide for the Care and Use of Laboratory Animals (NIH Publications No. 80-23, revised 1996) and approved by the Ethics Committee for Animal Research of the Fu Jen Catholic University (#A10848). This study included 42 rats in total. Fresh brain tissue from 30 rats was used to measure glutamate release, Ca^2+^ concentration, and membrane potential, as well as for HPLC, and Western blotting. Histological staining was performed on fixed brain tissue from nine rats.

### 4.2. Materials

Enmein was purchased from Tauto Biotech (Shanghai, China) at 98% purity. 4-AP, bafilomycin A1, GF109203X and Go6976 were purchased from Tocris Cookson (Bristol, UK). Fura-2-AM was purchased from Invitrogen (Carlsbad, CA, USA). ω-CgTX GVIA and ω-AgTX IVA were purchased from the Alomone lab (Jerusalem, Israel). PKC, PKC-α, MARCKS, EAAT3, synaptophysin, GFAP and β-actin antibodies were obtained from Cell Signaling (Beverly, MA, USA). EAAT2, PSD95, CD11b p-PKC and p-PKC-α, antibodies were obtained from Abcam (Cambridge, UK). NeuN antibody was obtained from Merck Millipore (Billerica, MA, USA). Horseradish peroxidase-conjugated secondary antibodies were obtained from Genetex (Zeeland, MI, USA). Alexa Fluor 488 and DyLight 594 were obtained from Invitrogen (Waltham, MA, USA). KA, DMSO, Cresyl violet, 4′,6-diamidino-2-phenylindole (DAPI) and general reagents were purchased from Sigma Chemical Co. (St Louis, MO, USA).

### 4.3. Preparation of Synaptosomes

Synaptosomes were prepared as previously reported [55,56]. Briefly, rats were killed by cervical dislocation and decapitation. The cerebral cortex was rapidly removed and homogenized in an ice-cold HEPES-buffered medium containing 0.32 M sucrose (pH 7.4). The homogenate was centrifuged at 3000× *g* for 10 min at 4 °C. The supernatant was retained and centrifuged at 14,500× *g* for 12 min at 4 °C. The pellet was resuspended and layered on top of a discontinuous Percoll gradient before being centrifuged at 32,500× *g* for 7 min at 4 °C. The protein concentration was determined using the Bradford assay. The synaptosomes were centrifuged in the final wash to obtain synaptosomal pellets containing 0.5 mg protein.

### 4.4. Determination of Glutamate Release

Synaptosomal pellets were analyzed for glutamate release using an enzyme-linked fluorescence method as previously described [55,57]. In brief, synaptosomes were suspended in HEPES-buffered medium containing 2 mM NADP^+^, 50 units of glutamate dehydrogenase, and 1.2 mM CaCl_2_, and the synaptosome suspension was stimulated with either 1 mM 4-AP or 15 mM KCl after 5 min. Fluorescence increases due to NADPH production were measured (λ excitation = 340 nm and λ emission = 460 nm) with a PerkinElmer LS55 spectrofluorimeter. The amount of glutamate released was calibrated by a standard of exogenous glutamate (5 nmol) and expressed as nanomoles glutamate per milligram synaptosomal protein (nmol/mg). The values given in the text and depicted in the bar graphs represent the levels of glutamate cumulatively released after 5 min of depolarization and are expressed as nmol/mg/5 min.

### 4.5. Determination of Exocytosis

Exocytosis was measured with an FM1-43 dye. FM1-43 is a fluorescent styryl dye that is lipophilic but not membrane permeable. When the styryl dye FM1-43 reversibly partitions into the outer leaflet of the exposed plasma membrane, its fluorescence increases. FM1-43 dye can be taken up into synaptic vesicles by endocytosis, and during subsequent exocytosis, FM1-43 is lost to the extracellular medium, resulting in a decrease in fluorescence [20]. Briefly, the synaptosomes were loaded with FM1-43 (100 μM) for 60 s before being loaded with extracellular high K^+^ (30 mM) for 90 s at room temperature. Then, the synaptosomes were washed with HEPES-buffered medium and centrifuged for 1 min at 3000× *g*, and the synaptosome suspension was stimulated with 1 mM 4-AP. The release of accumulated FM1-43 was measured as the decrease in fluorescence upon releasing dye into the solution (λ excitation = 488 nm and λ emission = 540 nm).

### 4.6. Determination of Cytosolic Free Ca^2+^ Concentration ([Ca^2+^]_C_)

[Ca^2+^]_C_ measurement was determined by the fluorescent probe Fura-2-AM. The synaptosomes were incubated with 5 μM Fura-2 in a HEPES-buffered medium containing 0.1 mM CaCl_2_ for 30 min at 37 °C. After Fura-2 loading, the synaptosomes were centrifuged for 1 min at 3000× *g,* and the pellets were resuspended in HEPES-buffered medium containing 1.2 mM CaCl_2_. Fura-2/Ca fluorescence was measured with a PerkinElmer LS55 spectrofluorimeter at 340 nm and 510 nm. [Ca^2+^]_C_ (nM) was calculated using the equations described previously [58].

### 4.7. Determination of Synaptosomal Membrane Potential

The synaptosomal membrane potential was detected using the positively charged membrane potential-sensitive carbocyanine dye DiSC_3_(5) [59]. DiSC_3_(5) fluorescence was measured (λ excitation = 646 nm and λ emission = 674 nm) with a PerkinElmer LS55 spectrofluorimeter. Data are expressed in fluorescence units.

### 4.8. Western Blot

Western blotting was performed as previously described [60]. Briefly, 30 μg of protein was loaded on a 10% sodium dodecyl sulfate (SDS)-polyacrylamide gel electrophoresis. After separation, the protein was blotted onto a nitrocellulose membrane and blocked for 1 h with 5% skim milk. The membrane was incubated with a specific primary antibody and a horseradish peroxidase-conjugated secondary antibody (1:5000) using enhanced chemiluminescence (ECL; Amersham, Buckinghamshire, UK). The primary antibodies used in this study included PKC (1:700), p-PKC (1:2000), PKC-α (1:600), p-PKC-α (1:2000), MARCKS (1:250), NeuN (1:5000), EAAT2 (1:50,000), EAAT3 (1:10,000), synaptophysin (1:50,000), PSD95 (1:1000), GFAP (1:5000), CD11b (1:9000), and β-actin (1:1000). Scanning densitometry was used to quantify four to five independent experiments, which were then analyzed with ImageJ software (Synoptics, Cambridge, UK).

### 4.9. Immunohistochemistry

The rats were divided into three groups at random: the DMSO-treated group (control), the KA-treated group, and the enmein 50 mg/kg + KA-treated group. Enmein was dissolved in a saline solution containing 1% DMSO and administered (i.p.) 30 min before KA injection (15 mg/kg in 0.9% NaCl, pH 7.0, i.p.). Seventy-two hours after KA injection, the rats were euthanized with CO_2_ and fixed by transcardial perfusion with 4% buffered paraformaldehyde (pH 7.2). The brains were removed and embedded in paraffin after fixation in 4% buffered paraformaldehyde, followed by 0.1 M phosphate-buffered saline (pH 7.2) for 24 h at 4 °C. A sliding microtome was used to acquire 25-μm-thick coronal brain sections from the hippocampus. For Nissl staining, the sections were mounted on a glass slide and stained with 0.1% aqueous Cresyl violet stain for 20 min after dehydration in graded ethanol solutions. For NeuN and GFAP staining, the sections were blocked in phosphate buffered saline containing 10% normal goat serum for 1 h to block nonspecific antibody binding before being incubated overnight at 4 °C with anti-NeuN antibody (1:500) or anti-GFAP antibody (1:1000) and subsequently incubated with IgG-DyLight 594 (1:1000) and FITC-conjugated IgG (1:1000) for 2 h at 25 °C. Nuclei were counterstained with DAPI (50 ng/mL). For CD11b staining, the sections were incubated overnight with anti-CD11b antibody (1:500). The sections were then incubated with goat biotinylated anti-mouse secondary antibodies (1:200; Vector Laboratories, Burlingame, CA, USA) for 2 h and subsequently incubated with ExtrAvidin peroxidase (1:1000) for 1 h at room temperature. After rinsing in 0.1 M phosphate-buffered saline for 20 min, the sections were reacted with 0.025% 3,3-diaminobenzidine tetrahydrochloride (DAB) solution in a phosphate-buffered saline containing 0.0025% hydrogen peroxide for 6 min. Immunostaining images were captured by a digital camera (Leica, Wetzlar, Germany) attached to a fluorescence microscope (Zeiss Axioskop 40, Göttingen, Lower Saxony, Germany). Leica 4×, 10× or 40× objective lenses with numerical apertures (NA) of 0.1 or 0.25 were used in this study. The number of surviving neurons and NeuN-, GFAP-, or CD11b-positive cells were counted in a 255 μm × 255 μm region of the hippocampal CA1/CA3 using ImageJ software (Synoptics, Cambridge, UK).

### 4.10. Glutamate Levels in Brain Tissue from High-Performance Liquid Chromatography (HPLC)

As previously reported [11], rats were killed by cervical dislocation and decapitation, and the hippocampus was rapidly dissected 72 h after KA injection. Fifty milligrams of hippocampal tissue were homogenized with 200 μL distilled water and centrifuged for 10 min at 1500× *g* (4 °C). The supernatant was filtered through 0.22 μm filters, and then 8 μL of the solution was injected into an HPLC system with electrochemical detection (HTEC-500, Eicom, Kyoto, Japan). The relative free glutamate concentration was determined based on peak areas by an external standard method.

### 4.11. Statistical Analysis

The mean ± standard error of the mean (SEM) was calculated. Statistical significance (*p* < 0.05) among groups was determined by Student’s *t* tests or one-way analysis of variance (ANOVA) followed by Tukey’s multiple comparisons test using GraphPad Prism software (La Jolla, CA, USA).

## 5. Conclusions

Our results show for the first time the glutamate release-inhibiting and anti-excitotoxic properties of enmein, a diterpenoid extracted from *Isodon japonicus* Hara. We demonstrate that enmein inhibits presynaptic glutamate release by suppressing Ca^2+^ influx and the PKC-dependent pathway. Furthermore, enmein prevents KA-induced glutamate excitotoxicity in vivo by suppressing gliosis and reducing glutamate accumulation via increased glutamate uptake. Our study may provide a pharmacological basis for the clinical use of enmein in developing new drugs for central nervous system diseases that involve glutamate excitotoxicity.

## Figures and Tables

**Figure 1 ijms-22-12966-f001:**
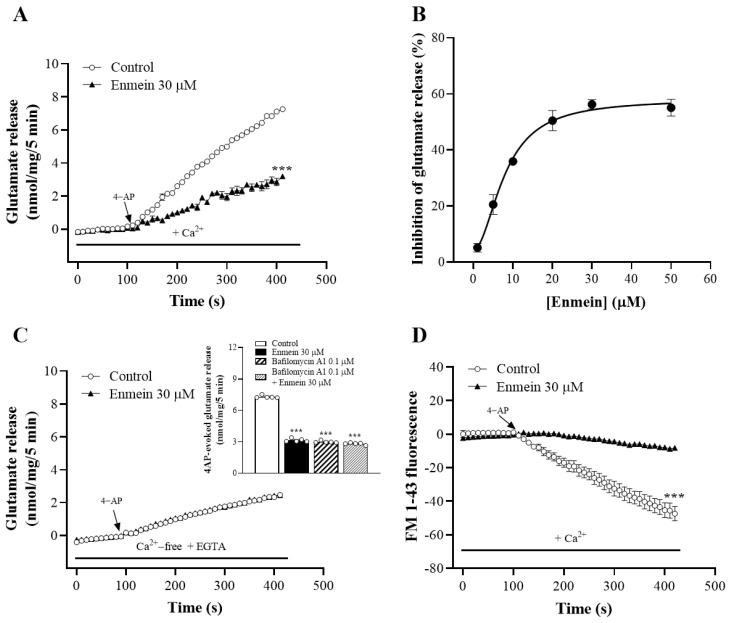
Effect of enmein on 4-AP-evoked glutamate release from rat cerebrocortical synaptosomes. (**A**) 4-AP (1 mM)-evoked glutamate release from synaptosomes incubated in the presence of 1.2 mM CaCl_2_, and in the absence (control) or presence of enmein. (**B**) Concentration–response curve of decreases in 4-AP-evoked glutamate release in the presence enmein. (**C**) Extracellular Ca^2+^-free solution containing EGTA (300 μM) on the action of enmein. Inset, glutamate release was evoked by 4-AP in the absence (control) or presence of bafilomycin A1, or bafilomycin A1 + enmein. (**D**) The release of FM1–43 was evoked by 4-AP in the absence (control) or presence of enmein. Enmein was added 10 min before the addition of 4-AP, and other drugs were added 10 min before this. Data are the mean ± SEM (*n* = 5 per group). ***, *p* < 0.001 vs. control group.

**Figure 2 ijms-22-12966-f002:**
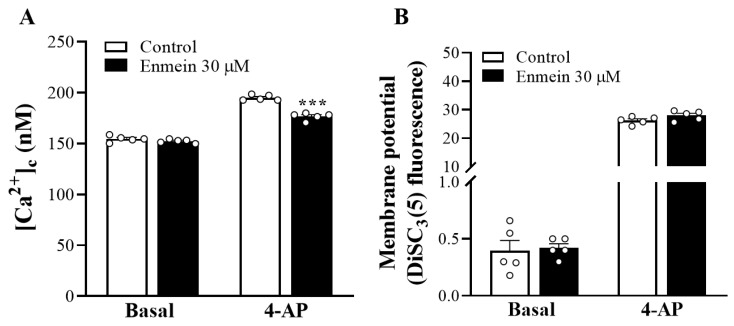
Effect of enmein on [Ca^2+^]_C_ (**A**) and the synaptosomal membrane potential (**B**). Enmein was added 10 min before the addition of 4-AP (1 mM). Data are presented as mean ± SEM. (*n* = 5 per group). ***, *p* < 0.001 vs. control group.

**Figure 3 ijms-22-12966-f003:**
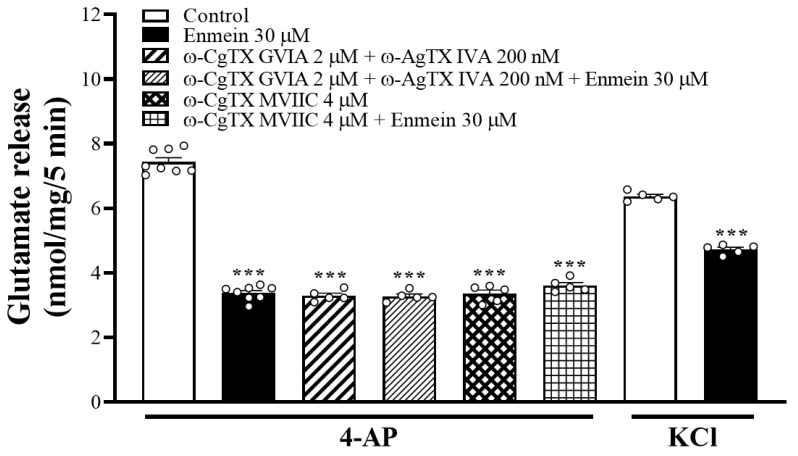
Blockade of N- and P/Q-type Ca^2+^ channels abolishes enmein inhibition of glutamate release. Glutamate release was evoked by 1 mM 4-AP in the absence (control) or presence of enmein, ω-CgTX GVIA + ω-AgTX IVA, or ω-CgTX GVIA + ω-AgTX IVA + enmein. Effect of enmein on the release of glutamate evoked by 15 mM KCl was also examined. Enmein was added 10 min before the addition of 4-AP, and other drugs were added 10 min before this. Data are presented as mean ± S.E.M. (*n* = 5 per group). ***, *p* < 0.001 vs. control group.

**Figure 4 ijms-22-12966-f004:**
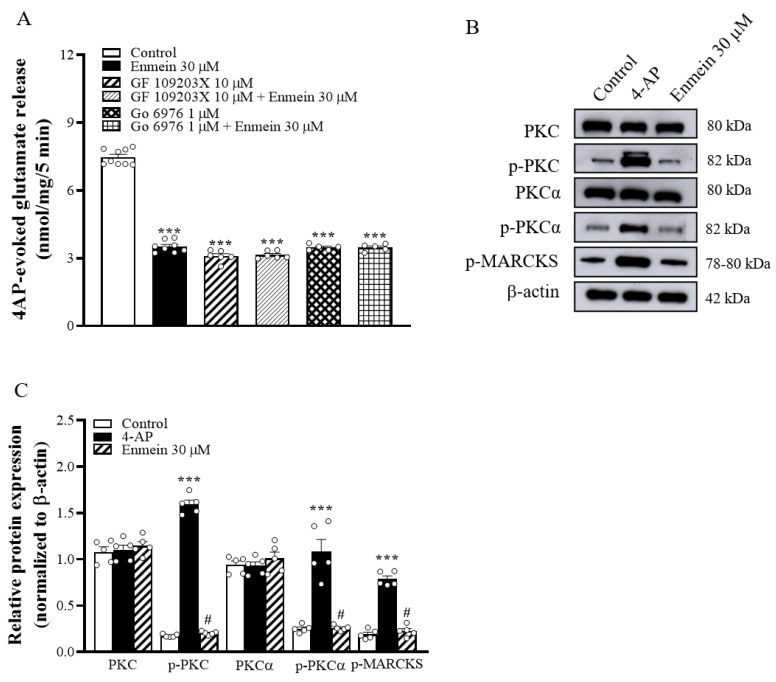
Enmein-mediated inhibition of glutamate release is linked to the suppression of PKC-dependent pathway. (**A**) Glutamate release was evoked by 1 mM 4-AP in the absence (control) or presence of enmein, GF109203X (a PKC inhibitor), Go6976 (a PKCα inhibitor), GF109203X + enmein, or Go6976 + enmein. (**B**) The expression of PKC, phospho-PKC (p-PKC), PKCα, phospho-PKC-α (p-PKC-α), and MARCKS was detected in synaptosomal lysates by Western blotting. (**C**) The quantification of protein expression of PKC, p-PKC, PKCα, p-PKCα, and MARCKS was normalized to β-actin. Enmein was added 10 min before the addition of 4-AP, and other drugs were added 20 min before this. Data are presented as mean ± SEM. (*n* = 5 per group). ***, *p* < 0.001 vs. control group. #, *p* < 0.05 vs. 4-AP group.

**Figure 5 ijms-22-12966-f005:**
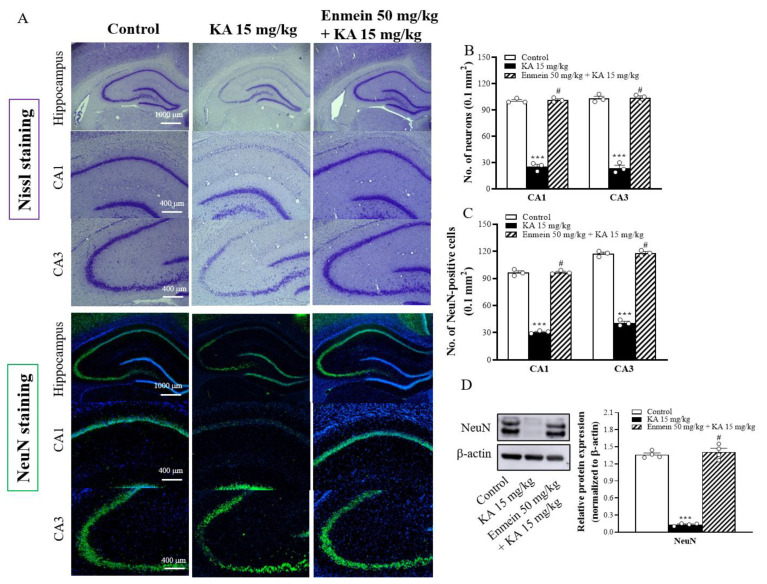
Effect of enmein pretreatment on the neuronal cell death in the hippocampus of rats with KA. (**A**) Representative images of Nissl and NeuN staining at 72 h after KA injection (i.p.). (**B**,**C**) Semiquantitative analysis of neurons and NeuN-positive cells in the CA1 and CA3 regions. (**D**) Western blot and semiquantitative analysis for NeuN in the hippocampus from differently treated groups. Data are presented as mean ± SEM. (*n* = 3–4 rats for each group). ***, *p* < 0.001 vs. control group. #, *p* < 0.05 vs. KA group.

**Figure 6 ijms-22-12966-f006:**
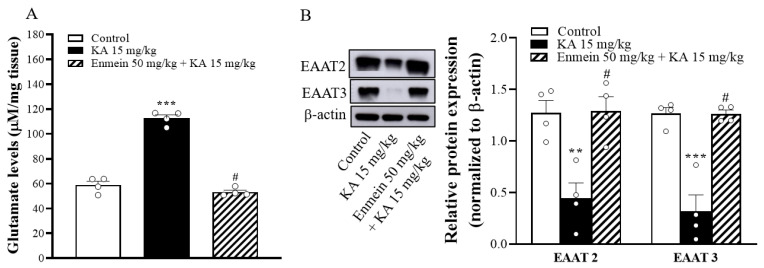
Effect of enmein pretreatment on the glutamate levels (**A**) and expression of EAAT2 and EAAT3 (**B**) in the hippocampus of rats with KA. The glutamate concentrations and EAAT2 and EAAT3 expression levels in the hippocampus were semiquantitative evaluated at 72 h after KA injection by high-performance liquid chromatography (HPLC) and Western blot, respectively. Data are presented as mean ± SEM. (*n* = 4 rats for each group). **, *p* < 0.01; ***, *p* < 0.001 vs. control group. #, *p* < 0.05 vs. KA group.

**Figure 7 ijms-22-12966-f007:**
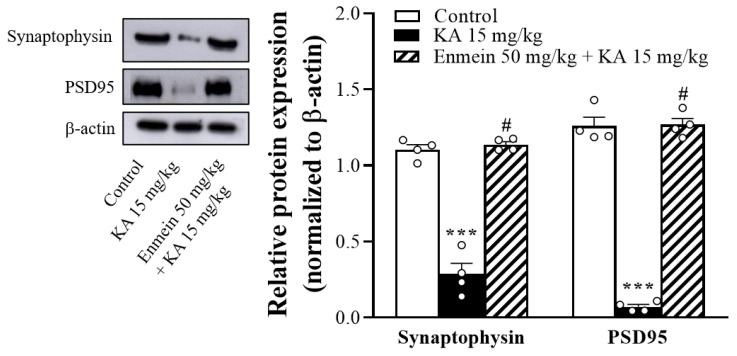
Effect of enmein on the expression levels of synaptophysin and PSD95 in the hippocampi of rats with KA. Western blot and semiquantitative analysis for PSD95 and synaptophysin in the hippocampus from differently treated groups. Data are presented as mean ± SEM. (*n* = 4 rats for each group). ***, *p* < 0.001 vs. control group. #, *p* < 0.05 vs. KA group.

**Figure 8 ijms-22-12966-f008:**
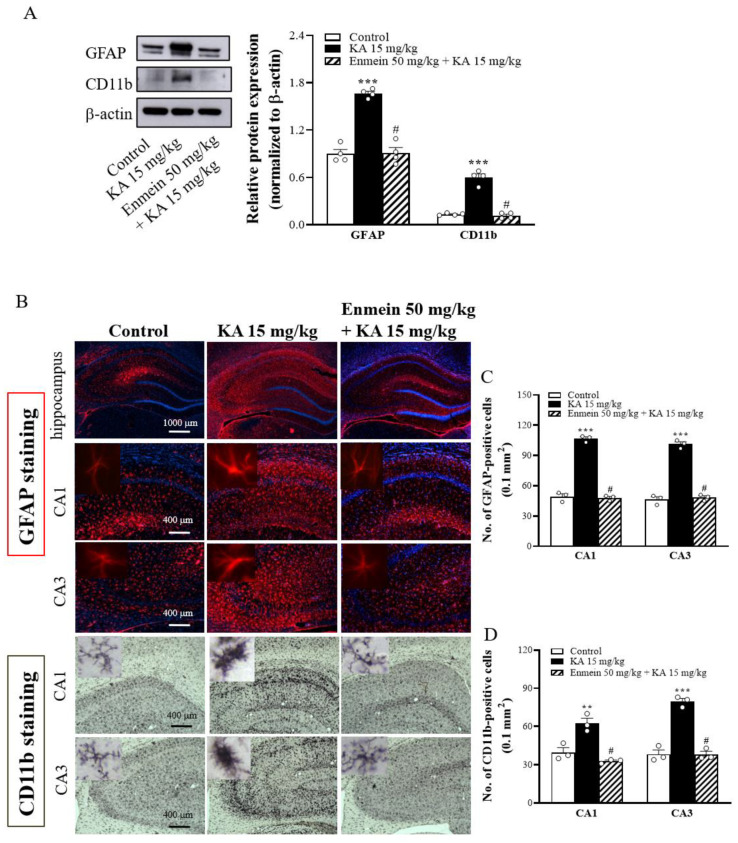
Effect of enmein on astrocyte and microglia activation in the hippocampus of rats with KA. (**A**) Western blot and semiquantitative analysis for the expression of GFAP and CD11b in the hippocampus at 72 h after KA injection. (**B**) Representative immunohistochemical mages of GFAP and CD11b staining at 72 h after i.p. KA. (**C**,**D**) Semiquantitative analysis of GFAP- and CD11b-positive neurons in the CA1 and CA3 regions. The insets are high magnification micrographs of astrocyte and microglial cells (Scale bar, 25 µm). Data are presented as mean ± SEM. (*n* = 4 rats for each group). **, *p* < 0.01; ***, *p* < 0.001 vs. control group. #, *p* < 0.05 vs. KA group.

## Data Availability

The data presented in this study are available on request from the corresponding author.

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
