# Peer review of "Enmein Decreases Synaptic Glutamate Release and Protects against Kainic Acid-Induced Brain Injury in Rats"

_ijms, 2021, doi:10.3390/ijms222312966_

Round 1
Reviewer 1 Report
In Abstract, Line 24-25 „In the KA 24 rat model, intraperitoneal administration of enmein 30 min before intraperitoneal injection of KA“, please, insert this part of the sentence in the very beginning of the description of the Results.
Replace “living” for the number of neurons in the text to Fig. 5 B,C with “neurons”.
Is n=3 (number of rats) for histological analysis enough number for quantitative estimation? The same question is for WB and HPLC analysis (Fig. Fig. 6A, 6B). In my opinion, for quantitative evaluation of drug effects it is necessary for histological data to use at least n=4 rats and for HPLC at least n=6 rats. Although WB is used mostly to confirm other methods (as is the case with HPLC) also at least four rats are needed for evaluation.
Also, in the text to Fig. 6A,B, please, put the two methods HPLC and WB in the same order as they are presented in the figures above the text.
Line 468 replace crystal: with Cresyl violet.
Line 488 (4.4.) what type of anesthesia was applied before decapitation? Please, describe haw were decapitated the rats. I guess via guillotine.
The Conclusion needs revision. The first two sentences are part of the Introduction or Discussion. The third sentence (Our results show 503 the glutamate release-inhibiting and anti-excitotoxic properties….) also need an edition.

Author Response
Response to reviewer1
ijms-1476660R1
We thank the reviewer for the critical comments and constructive suggestions.
In Abstract, Line 24-25 „In the KA 24 rat model, intraperitoneal administration of enmein 30 min before intraperitoneal injection of KA“, please, insert this part of the sentence in the very beginning of the description of the Results.
According to this point, the sentences〝rats were given either enmein or dimethylsulfoxide (DMSO) intraperitoneally (i.p.) 30 min before KA injection (i.p.).〞is included in the result section (Page 7, Line 179-180).
Replace “living” for the number of neurons in the text to Fig. 5 B, C with “neurons”.
As suggestion by the reviewer, in the figure 5B, the word is changed to neurons.
Is n=3 (number of rats) for histological analysis enough number for quantitative estimation? The same question is for WB and HPLC analysis (Fig. Fig. 6A, 6B). In my opinion, for quantitative evaluation of drug effects it is necessary for histological data to use at least n=4 rats and for HPLC at least n=6 rats. Although WB is used mostly to confirm other methods (as is the case with HPLC) also at least four rats are needed for evaluation.
This study was designed with two goals in mind: (1) minimizing animal suffering and (2) using as few animals as possible (Page 17, Lines 391-392). In the present in vivo study, histological staining was performed on fixed brain tissue from 9 rats (n =3 for each group), and fresh brain tissue from 12 rats was used for HPLC and Western blotting (n =4 for each group). However, the quantitative results are statistically significant and consistent.
Also, in the text to Fig. 6A, B, please, put the two methods HPLC and WB in the same order as they are presented in the figures above the text.
As suggestion by the reviewer, the sentence is modified to 〝To elucidate the possible mechanism of the neuroprotective action of enmein in our study, glutamate concentrations—a key neurotransmitter in the central nervous system underlying the pathophysiology of neurological disorders—were determined 72 h after KA injection by high-performance liquid chromatography (HPLC).〞(Page 10, Line 205-208), and 〝As shown in Figure 6B, Western blot analysis showed that EAAT2 and EAAT3 hippocampal levels were significantly lower in the KA-treated group than in the control group〞(Page 10, Line 216-218).
Line 468 replace crystal: with Cresyl violet.
The word is changed to Cresyl (Page 17, Line 410).
Line 488 (4.4.) what type of anesthesia was applied before decapitation? Please, describe haw were decapitated the rats. I guess via guillotine.
As suggestion by the reviewer, the sentence is modified to〝rats were killed by cervical dislocation and decapitation〞(page 19, Line 502-503).
The Conclusion needs revision. The first two sentences are part of the Introduction or Discussion. The third sentence (Our results show 503 the glutamate release-inhibiting and anti-excitotoxic properties….) also need an edition.
As suggestion by the reviewer, the sentences are modified to 〝Our results show the glutamate release-inhibiting and anti-excitotoxic properties of enmein, a diterpenoid extracted from Isodon japonicus Hara, for the first time. We demonstrate that enmein inhibits presynaptic glutamate release by suppressing Ca2+ influx and the PKC-dependent pathway, and enmein prevents KA-induced glutamate excitotoxicity in vivo by suppressing gliosis and reducing glutamate accumulation via increased glutamate uptake. Our study may provide a pharmacological basis for the clinical use of enmein in developing new drugs for central nervous system diseases that involve glutamate excitotoxicity.〞(Page 19, Lines 515-526).

Reviewer 2 Report
This manuscript deals that enmein, compound derived from Isodon japonicus ameliorates depolarization-evoked synaptic glutamate release by modulating Ca2+ influx and PKC, and protects hippocampal neurons from kainic acid-induced neurotoxicity via suppressing glial and microglial activation and accelerating glutamate uptake in rats. Submitted results consider as very convincible and persuadable scientifically. Accordingly, this manuscript seems enough acceptable in this journal after minor revision as follows;
- In Fig. 8, it is difficult to identify immunostaining figures clearly. Accordingly, authors should add more magnified pictures in GFAP and CD11b staining results.
Author Response
Response to reviewer2
ijms-1476660R1
We thank the reviewer for the critical comments and constructive suggestions.
- In Fig. 8, it is difficult to identify immunostaining figures clearly. Accordingly, authors should add more magnified pictures in GFAP and CD11b staining results.
As suggestion by the reviewer, the quality of GFAP and CD11b staining images (Figure 8) is improved.

Round 2
Reviewer 1 Report
The authors have satisfactorily addressed most of my comments. However, there are still some points needed clarification.
As concerns the number of rats used per group, the authors responded that: “This study was designed with two goals in mind: (1) minimizing animal suffering and (2) using as few animals as possible (Page 17, Lines 391-392)”. The study design must satisfy both ethical issues and sufficient statistical data to prove the hypothesis. Therefore, it is not enough for a valid hypothesis to minimize the number of rats only. I disagree that n = 4 (HPLC or WB) or n = 3 (histology) rats per group is a quantitative analysis. Because the authors used three different methods, I will accept the suggested design with minimal animals. However, please indicate that the investigation is semi-quantitative.
For the method of decapitation, the authors responded that〝rats were killed by cervical dislocation and decapitation〞(page 19, Line 502-503). Are the rats killed without anesthesia? I don’t think that killing via cervical dislocation is an approach of “minimizing animal suffering as the authors claimed above. Also, cervical dislocation leads to exitus, and applying decapitation after that is unnecessary.
The revised conclusion needs an additional minor edition. In the first sentence, “Our results show the glutamate release-inhibiting and anti-excitotoxic properties of enmein, a diterpenoid extracted from Isodon japonicus Hara, for the first time” move “for the first time” after “Our results show”. The next sentence is too long and split in two separate sentences.

Author Response
Response to reviewer1
ijms-1476660R2
We thank the reviewer for the critical comments and constructive suggestions.
As concerns the number of rats used per group, the authors responded that: “This study was designed with two goals in mind: (1) minimizing animal suffering and (2) using as few animals as possible (Page 17, Lines 391-392)”. The study design must satisfy both ethical issues and sufficient statistical data to prove the hypothesis. Therefore, it is not enough for a valid hypothesis to minimize the number of rats only. I disagree that n = 4 (HPLC or WB) or n = 3 (histology) rats per group is a quantitative analysis. Because the authors used three different methods, I will accept the suggested design with minimal animals. However, please indicate that the investigation is semi-quantitative.
As suggestion by the reviewer, the sentence is modified to 〝All experimental protocols were designed with minimal animals.〞(Page 17, Line 394-395 ). In addition, the sentences are modified to 〝Semiquantitative analysis of neurons and NeuN-positive cells in the CA1 and CA3 regions. (D) Western blot and semiquantitative analysis for NeuN in the hippocampus from differently treated groups.〞(Page 10, Lines 200-202); 〝The glutamate concentrations and EAAT2 and EAAT3 expression levels in the hippocampus were semiquantitative evaluated at 72 h after KA injection by high-performance liquid chromatography (HPLC) and western blot, respectively (Page 11, Lines 227); Western blot and semiquantitative analysis for PSD95 and synaptophysin in the hippocampus from differently treated groups (Page 12, Lines 241); Western blot and semiquantitative analysis for the expression of GFAP and CD11b in the hippocampus at 72 h after KA injection. (B) Representative immunohistochemical mages of GFAP and CD11b staining at 72 h after i.p. KA. (C, D) Semiquantitative analysis of GFAP- and CD11b-positive neurons in the CA1 and CA3 regions (Page 14, Lines 266-269)〞.
For the method of decapitation, the authors responded that〝rats were killed by cervical dislocation and decapitation〞(page 19, Line 502-503). Are the rats killed without anesthesia? I don’t think that killing via cervical dislocation is an approach of “minimizing animal suffering as the authors claimed above. Also, cervical dislocation leads to exitus, and applying decapitation after that is unnecessary.
As suggestion by the reviewer, the sentence is modified to 〝All experimental protocols were designed with minimal animals.〞(Page 17, Line 394-395).
The revised conclusion needs an additional minor edition. In the first sentence, “Our results show the glutamate release-inhibiting and anti-excitotoxic properties of enmein, a diterpenoid extracted from Isodon japonicus Hara, for the first time” move “for the first time” after “Our results show”. The next sentence is too long and split in two separate sentences.
As suggestion by the reviewer, the sentence is modified to 〝Our results show for the first time the glutamate release-inhibiting and anti-excitotoxic properties of enmein, aditerpenoid extracted from Isodon japonicus Hara. We demonstrate that enmein inhibits presynaptic glutamate release by suppressing Ca2+ influx and the PKC-dependent pathway. Furthermore, enmein prevents KA-induced glutamate excitotoxicity in vivo by suppressing gliosis and reducing glutamate accumulation via increased glutamate uptake..〞(Page 19, Line 520-527 ).
